

# First wild boar density data from Araucaria forest in Patagonian Andes

Oscar Skewes[1,*], Annaluisa Kambas[2,*], Paula Gädicke[1] and Oliver Keuling[2]

[1] Facultad Ciencias Veterinarias, Universidad de Concepción, Chillán, Chile
[2] Institute for Terrestrial and Aquatic Wildlife Research, University of Veterinary Medicine Hannover, Hannover, Germany
[*] These authors contributed equally to this work.

## ABSTRACT

As *Sus scrofa* is an invasive species in South America, it may have a significant impact on biodiversity. Evaluating this threat requires reliable data, and population density can serve as a critical measure. However, such data is currently lacking for the southern Andes region. To address this gap, we monitored wild boar density in the Villarrica National Park, located in the Andes of south-central Chile. This study area is notable not only for its challenging climatic conditions but also for its endangered *Araucaria araucana* forest, which provides abundant food resources during autumn seed fall. The density calculated for the entire study period was 1.4 individuals/km$^2$, with no significant variation between cold and warm seasons. The encounter rate showed strongly monthly variations. Given that this represents the first density estimate for wild boar in this region, our findings emphasize the need for continued monitoring, particularly due to the potential threat to the ecosystem and the already endangered Araucaria forest.

## INTRODUCTION

The wild boar (*Sus scrofa*) is an invasive species in South America and was first introduced there in 1904 in Argentina with animals from Europe. They were soon relocated to different parts of the country including the southern Andes. A contingent also arrived directly in Chile many years later, and the first population of wild boars in Chile likely existed around 1950 and is attributed to the direct import of some animals from Germany (*Skewes & Jaksic, 2015*). The main reason for the release of these animals was for hunting, but some individuals also escaped from farms by accident. The Argentinian and Chilean populations intermixed and formed the basis of the present wild boar population of the southern cone of America (*Cuevas et al., 2021*).

Wild boars can have detrimental effects on biodiversity, agriculture, and livestock (*Barrios-Garcia & Ballari, 2012*). These effects are often density-dependent, meaning that higher population densities lead to more severe impacts on biodiversity, greater agricultural damage, and higher disease risks (*Fulgione & Buglione, 2022*). They are described as ecosystem engineers, as they can significantly impact habitats. This occurs in their native

Corresponding author
Oliver Keuling, oliver.keuling@tiho-hannover.de

habitats (*Croft et al., 2020*) as well in non-native environments (*Risch et al., 2010*). Given that the severity of their impact correlates strongly with population density, assessing the density of wild boar populations is crucial for understanding and mitigating these effects. Density assessments can be achieved through methods such as the Random Encounter Model (REM), which estimates population density based on the frequency of animal encounters (*Rowcliffe et al., 2008*); distance sampling, which involves measuring the distances of observed animals from a line or point to estimate density (*Thomas et al., 2010*); and capture-recapture techniques, where animals are captured, marked, and released to determine population size based on subsequent recapture rates (*Otis et al., 1978*). These approaches provide essential data to inform management strategies aimed at reducing the negative impacts of wild boar on ecosystems and agricultural areas.

In 2014, the first outbreaks of African Swine Fever, affecting wild boar in the European Union, increased public concern (*Jori et al., 2021*). Following, experts of the EU under the ENETWILD Consortium selected a method to estimate wild boar density (*Enetwild-Consortium et al., 2018*). As a result, they adopt the procedure based on images captured by camera traps and processed with the REM (*Rowcliffe et al., 2008*; *Palencia et al., 2022*). Thus, wild boar density values in Europe ranged from 0.35 individuals/km$^2$ to 15.25 individuals/km$^2$ (*Enetwild-Consortium et al., 2022*). But it is exactly this varying range of density in Europe that shows the impact of different environments on the species.

Therefore, it is necessary to have data about the density of this species, to evaluate and possibly take further management actions. However, there is no information on the density of wild boar in Chile, even though it extends to important habitat for many threatened and endangered species and is also considered one of the 36 Biodiversity Hotspots in the world (*Myers et al., 2000*). Consequently, we studied the population density of wild boar in an Araucaria (*Araucaria araucana)* forest.

Notably, *A. araucana* is a Gondwana long-lived coniferous and endangered species with a small range (*Premoli, Quiroga & Gardner, 2017*). It is not only important for conservation value, but also because the seeds are collected by the indigenous groups of Chile and Argentina and also serve as food for native birds and rodents (*Sanguinetti et al., 2023*). The species was classified as endangered by the IUCN in 2013, which was mainly caused by fires, years of logging and invasive species. Although wild boar has been present in these forests for decades (*Skewes & Jaksic, 2015*), the impact of wild boar as depredator of seed may shift from individual trees to stand scale, threatening Araucaria forest regeneration (*Sanguinetti & Kitzberger, 2010*).

Considering the multiple impacts of an enlarged wild boar population, investigating the population density is important for estimating this danger for humans and nature, especially for the endangered Araucaria. This study aims to provide, for the first time, an estimate of wild boar density in Araucaria forests. Based on these results, conservation actions for this endangered forest species and wild boar management measures can be proposed in the future.

## MATERIALS AND METHODS

### Study area

The study site named "Puesco" is in the Andes of south-central Chile (39°35′S, 71°31′W) at elevations of 1,200 to 1,400 m asl and covers 15 km². It is located on the north side of the Lanín volcano in the Villarrica National Park (permission no. 03/2015 CONAF) in the Araucanía Region, Chile (Fig. 1).

There is no hunting, visitors circulate along trails and in autumn the neighboring indigenous communities collect Araucaria seeds. According to the climatic station "Ea. Mamuil-Malal" (39°64′739″S, 71°26′955″W) at 900 m asl and circa 20 km east, the mean annual precipitation is 1,081 mm and the mean annual temperature is 9.3 °C. The coldest month is July with a mean of 1.3 °C and the warmest is February with 15.5 °C (http://www.aic.gov.ar/sitio/estaciones). In the site snow falls from June to September, the snow cover stays for approximately 45 days with a maximum height of up to 0.9 m (Author observation).

The site is dominated by forests of the long-lived monkey puzzle tree *A. araucana* mixed with lenga beech *Nothofagus pumilio*. The understory is composed of poaces *Festuca gracillima*, *Alstroemeria aurea*, and patches of dense bamboo *Chusquea* spp. thickets, *Gaultheria* sp. and *Nothofagus antarctica* shrubs (<5 m height). The Araucaria seed fall starts in March and ends in June, but seeds are also available in spring after the thick snow cover melts. As there are no other fruit bearing trees in the study area and the other occurring plants are not as remotely comparable nutritious, the food availability changes drastically over the year.

### Data capture

We conducted a camera trap (CT) study from May 2020 to April 2022. It was carried out by the deployment of 10 Ltl Acorn® 6210 CTs, that have a trigger speed of 0.8 s. The location of the CTs were set randomly on the Google Earth® platform with a 1,000 to 1,400 m distance between each CT on the map. At each site, an area with 10 m of clear vision in front of the camera lens was selected. CTs were attached to a tree (diameter > 20 cm) at 1 m above ground, facing north or south, to prevent the sun flare from the sunrise or sunset that results in overexposed photos where animals become challenging to identify (*Apps & McNutt, 2018*). Even though *Palencia et al. (2021)* suggested that the height level should be at shoulder height, which would be 80 cm in case of the wild boar, we had to attach the cameras at 1 m height, as there can be up to 0.9 m snow in winter (author observation, compare Fig. 2).

Despite two CTs being moved from the initial place, maintaining the restrictions already described. CTs were programmed to capture three consecutive images, with no delay and with normal PIR sensitivity. We did not use bait or attractors at the site. The CTs were maintained every 3 to 5 months, depending on the weather conditions.

This CT study was authorized by the national forestry corporation CONAF in Chile. Field permit to conduct the study in the National Park of Villarrica was given by Ministerio de Agricultura (Chile), Depto. Areas Silvestres Protegidas Region de la Araucaria. The Bioethics commission of the University of Concepcion approved the study. The pictures of
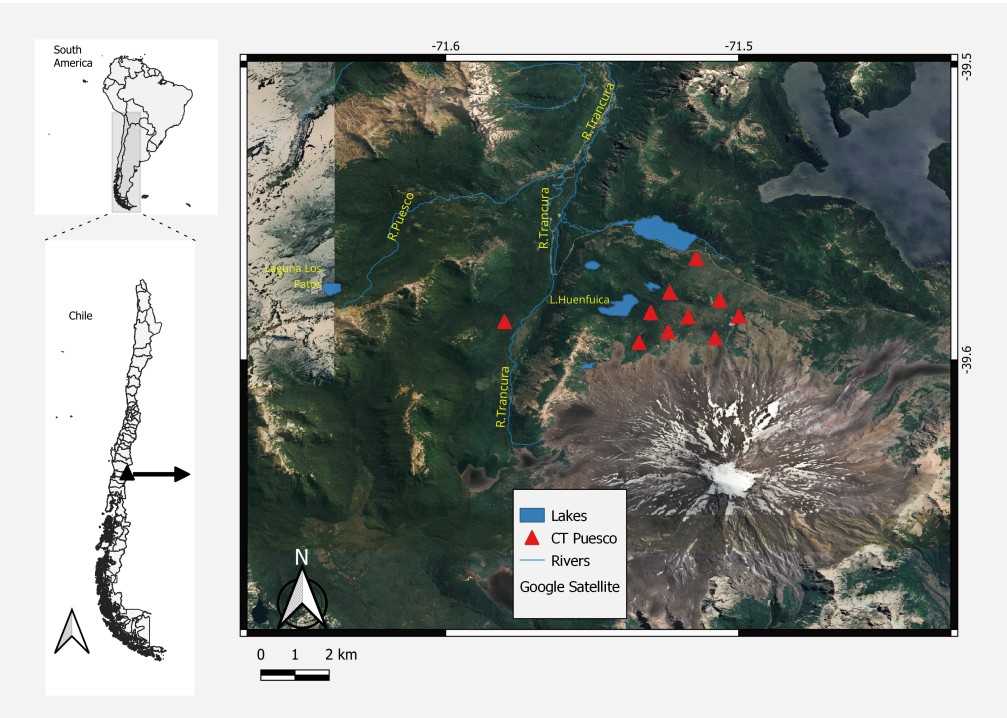

**Figure 1 Map of the study area.** Map created with Free and Open Source QGIS (http://qgis.org), under a CC BY-SA license.

people were handled following the guidelines of *Sharma et al. (2020)*, which means that the privacy of individuals inadvertently photographed by camera traps was strictly protected. Photos of unknown people were securely stored and not disclosed. Suspected researchers in photos were consulted on whether to destroy or receive the images. Wildlife images were shared with and credited to the relevant agencies overseeing the National Park, ensuring proper use and acknowledgment.

## Data processing

All photographs were screened by the authors to identify those that contained wild boar (compare Fig. 2). The REM (Random Encounter Model) by *Palencia et al. (2021)* estimates animal densities using CT data. It models random encounters between animals and cameras, factoring in animal movement and detection probability. For the REM, we used all wild boar images, irrespective of the time between pictures. Density is calculated using the following formula:

$$D = \frac{y}{t} \times \frac{\pi}{v \times r \times (2 + \theta)} \tag{1}$$

where $\frac{y}{t}$ denotes the trapping rate, the number of captures per unit time, $v$ is the average speed of the animals, $r$ is the effective detection range of the camera, and $\theta$ is the camera's detection angle.

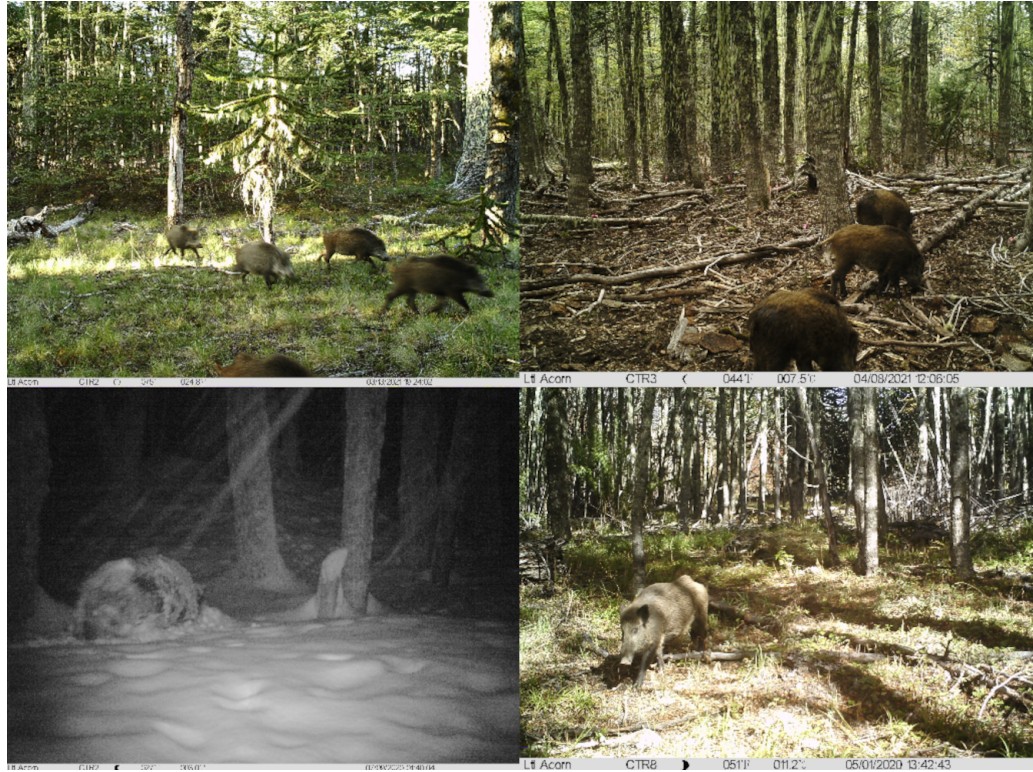

**Figure 2** Pictures of wild boars captured by CTs in the study area.

We divided the study period into cold and warm seasons. The cold season spanned from May to October, while the warm season covered the months from November to April.

Activity level was estimated after *Rowcliffe et al. (2014)* using the R package 'activity' (*Rowcliffe, 2023*). The REM density was calculated for all CTs and then averaged for season. The activity level between seasons was tested with 1,000 bootstrap replications. To establish the angle and radius of detection, we carried out thorough walk tests at our CTs (*Cusack et al., 2015*).

Day range, which is the average daily distance traveled by an individual was calculated based on speed and activity level (*Palencia et al., 2019*). The speed of the wild boars was obtained from information from CTs and processed as described by *Rowcliffe et al. (2016)*. In this method, we divided the distance traveled by the duration of the sequence (the difference in time between the timestamps on the first and last picture). To the end of the study, we recorded the location of every captured wild boar in one single image for each CT. Then, in the field, with the diagram in hand, we measured the corresponding locations with tape. Subsequently, the CT images were reviewed, and the distance traveled, as well as time were noted for each animal sequence. Those sequences in which animals reacted to the CT or in which there was only one image of wild boar were considered for encounter rate but not for speed (*Rowcliffe et al., 2016*). The encounter rate is the number of (n°) contacts divided by the product of n° CTs and n° days. For the monthly encounter

rate, we considered independent images as a 30 min interval of the same CT. To analyze differences in monthly encounter rates, the Chi-square test for independence was carried out with 95% confidence level. Also, the Kruskal-Wallis's test was carried out.

All the statistical analyses were performed on Infostat software (*Di Renzo et al., 2020*).

## RESULTS

The total effort for this study involved 4,703 24h-periods, with 2,516 in the cold season and 2,187 in the warm season. We had a total number of wild boar encounters of 370, of which 320 were independent. Our CT had a detection angle of 0.741 radians and a detection radius of 8.0 m. From analyzing 280 image sequences, we estimated the speed of movements to be 0.43, 0.49, and 0.42 m/s for the two years, cold and warm seasons, respectively. We found significant differences in speed between the seasons (*p* value = 0.0334, Kruskal-Wallis's test).

In terms of encounter rates, there are significant differences among months. March and April had significantly the highest rates at 0.24 and 0.25, respectively (Chi square, *df* = 11, *p* < 0.05). The lowest encounter rates were in September with no pictures at all during the study and in August with only 0.005 (two pictures) (*p* < 0.05) (Fig. 3).

The activity index was determined by analyzing 370 pictures. The overall activity index was calculated to be 0.48 (SE ± 0.05). During the cold season, the activity index was 0.45 (SE ± 0.06), while during the warm season it was 0.43 (SE ± 0.03) (Table 1). The overall group size was 2.1 (SE ± 0.1), with a group size of 2.3 (SE ± 0.4) in the cold season and 2.0 (SE ± 0.2) in the warm season (Table 2). There was no significant difference in the activity between seasons (*p* > 0.05).

Using the activity index and estimated animal speed, we were able to determine that the animals had a day range of 11.4 km over two years. Accordingly, throughout the entire study period, the estimated population density was 1.4 individuals/km$^2$ (Table 2). In the cold season, the density was 1.0 individuals/km$^2$, while in the warm season it was 2.6 individuals/km$^2$.

## DISCUSSION

While interpreting our results, it is crucial to acknowledge certain limitations. Firstly, the relatively small number of CTs used was due to budget constraints. Surveys range from 1 to 1,000 CTs (*Burton et al., 2015*). Typically, 20 to 30 CTs are recommended for monitoring populations of medium to large mammals, such as wild boars, to ensure sufficient detection and data collection across a given study area (*Kays et al., 2020*). Our small sample size likely caused wide confidence intervals for density. The variability in CT captures may be due to random placement or the small number of CTs.

*Massei et al. (2018)* suggested a minimum of nine cameras per km$^2$ for evaluating wild boar density, while our study had a much lower density (0.7 cameras per km$^2$). According to *Guerrasio et al. (2022)*, higher camera density would reduce error related to the contact rate. Thus, our data should be interpreted cautiously due to high confidence intervals. The nested CI analysis showed that adding more CTs decreases CI width without stabilization,
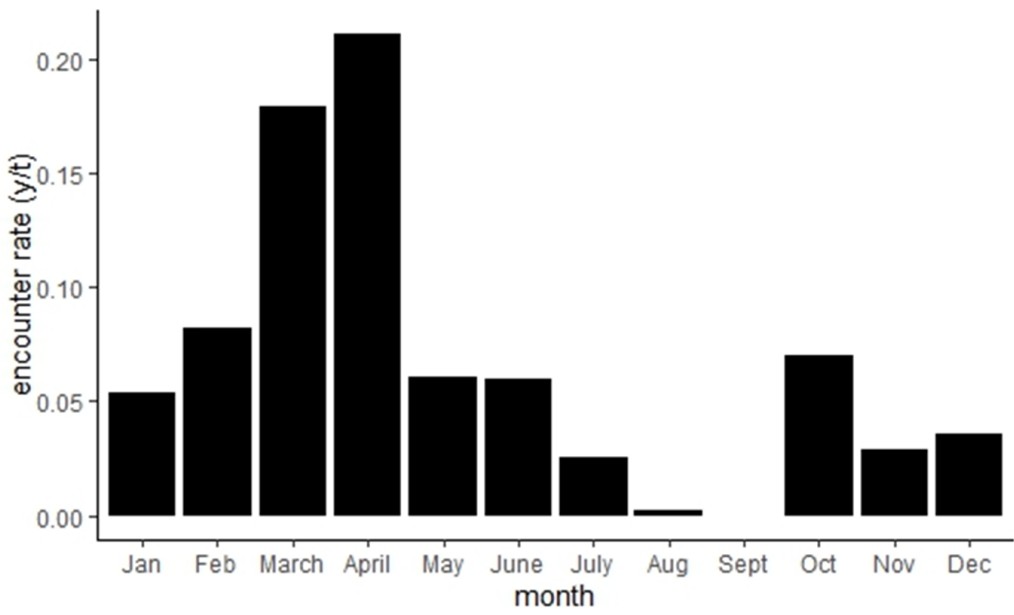

**Figure 3** Monthly encounter rate (encounter ($y$)/ time ($t$)) of wild boar in Villarrica NP 177 (Chile) from May 2020 until April 2022 ($n = 370$).

**Table 1 Estimated activity index for SE wild boar in study area.**

|           | Entire period | Cold season | Warm season |
|-----------|---------------|-------------|-------------|
| act       | 0.48          | 0.45        | 0.43        |
| Se        | 0.05          | 0.06        | 0.03        |
| lcl 2.5%  | 0.43          | 0.31        | 0.33        |
| ucl 97.5% | 0.56          | 0.49        | 0.44        |

**Notes.**

act, activity index Activity package Rowcliffe 2023; lcl 2.5%, lower confidence limit; ucl 97.5%, upper confidence limit.

indicating the need for more CTs for accurate results. Overestimating precision could undermine management practices (*Guerrasio et al., 2022*).

Second, the trigger speed of our cameras was slower than the 0.5 s recommended by *Palencia et al. (2021)*, operating at 0.8 s, which may have impacted detection probability. Additionally, our CTs were positioned 20 cm higher than *Palencia et al.*'s *(2021)* recommendation due to significant snow accumulation in winter, a factor that could have also influenced detection probability. However, the CTs used to have detection probabilities near 1 for wild boar similar to other models, such as Bushnell, (*Palencia et al., 2021*), when employed in the same way. Our CTs, still almost covered by snow, continued to capture images, for example, of *Lepus europaeus* during the harsh winter.

To evaluate the meaning of our results, it is essential to compare our findings with previous studies. Our study's unique contribution is the quantification of population density, which has not been investigated in prior research in this geographical area. Most previous studies have focused on estimating population abundance, but our research

**Table 2 Estimated random encounter model (REM) parameter values for each period.** Where $y/t$ is the encounter rate (nº contacts/nº camera traps*days); $v$, the average distance travelled by an individual during a day (day range); $r$, the radius of detection. We present standard errors (SE), 95% confidence intervals and coefficient of variation (CV, %) for density.

| Season | Entire period | Cold season | Warm season |
|---|---|---|---|
| $y/t$ (ind/CT day) | 0.07 (370/4,703) | 0.035 (88/2,516) | 0.128 (282/2,181) |
| $v$ (km/day) | 11.4 | 11.6 | 10.0 |
| $r$ (km) | 0.008 | 0.008 | 0.008 |
| Group size ($\pm$SE) | 2.1 (0.1) | 2.3 (0.4) | 2.0 (0.2) |
| Density (indiv/km$^2$) ($\pm$SE) | 1.4 (0.6) | 1.0 (0.5) | 2.6 (1.0) |
| low IC 95% (indiv/km$^2$) | 0.36 | 0.1 | 0.6 |
| Upper IC 95% (indiv/km$^2$) | 2.6 | 1.9 | 4.6 |
| CV (%) | 128.6 | 145.0 | 124.1 |

provides an added dimension by considering the seasonal variation in movement speed, activity index and therefore density, thereby enriching the understanding of animal behavior in different climatic conditions.

The estimated density of 1.4 individuals/km$^2$ is the first ever calculated density for wild boar in South American temperate forests. However, our density is still difficult to evaluate. Compared to the European average density of 7.8 individuals/km$^2$ (*Guerrasio et al., 2022*), our density may seem rather low. This is also considering that there is no hunting in the area and the availability of the nutritious seeds of the Araucaria, although the presence of puma can have some effect on the daily movement, as they hunt the wild boar as well (*Skewes et al., 2012*).

This low density can presumably not be explained by the later introduction of the species in South America, as the species is highly adaptable and reproductive. The first introduction in this area was 70 years ago (*Skewes & Jaksic, 2015*), the population could be at a higher level just by reproductive rate. The different environmental factors are surely causing this disruption.

When comparing the European data to ours, it must be taken into account that most of the studies have not been conducted in extreme ecosystems, but in ones that are native to the wild boar and in areas of known wild boar abundances. This difference includes the low food availability at the study site, which is caused by the rough environmental conditions at 1,400 m altitude. We suggest that the reason for April being the month with the highest encounter rate is on the one hand, that it is the rutting time of wild boar in Chile (*Skewes, 1990*), which causes a higher mobility of animals (*Morelle et al., 2015*). On the other hand, because in March it is the Araucaria seed fall, which leads to higher food availability (*Sanguinetti & Kitzberger, 2008*). The Araucaria seeds are not only consumed by animals like wild boars and rodents, but also by the indigenous people that collect the seeds. The main component of the 3–4 g weight Araucaria seeds is starch (about 88.0 g/100 g solids) (*Henríquez et al., 2008*) followed by protein (about 7.0 g/100 g solids). The protein of this seed has a high nutritional quality, like that of soy protein

(*Conforti & Lupano, 2011*). Also, the Araucaria seeds as a food item for wild boar has been described (*Pelliza-Sbriller & Borrelli, 2008*).

The fluctuation between the cold and warm seasons should be considered. To further evaluate them, the climatic conditions need to be analyzed. Snow is present from June to September, including July being overall the coldest month with an average of 1.3 °C (http://www.aic.gov.ar/sitio/estaciones). The changes in temperature in the cold season not only generate a drastic decrease in edible flora, but also impede movement through up to 90 cm of snow. Even though wild boars decrease their activity in winter, the high snow layer hinders them from any longer movement in these months. These conditions, combined with the significantly lower encounter rate, suggest that the population could be moving between areas seasonally. This would be consistent with findings from research conducted in Poland, as well as in mountainous regions in Italy and Spain (*Andrzejewski & Jezierski, 1978*; *D'Andrea et al., 1995*; *Sarasa & Sarasa, 2013*). In consequence, it could be hypothesized that wild boars migrate to the Araucaria forest due to the increased food availability. This would pose a risk to the endangered Araucaria trees, as the boars' chewing could hinder seed germination. In Europe (*Jezek et al., 2021*), the damaging of seedlings through wild boars has been described, which could be possible with the Araucaria seedlings as well. This would imply an even higher damage to the already endangered species.

Considering the limitations to our study, we propose a continuing survey of the wild boar population with more CTs and possibly even management actions, if further population growth is observed.

## CONCLUSIONS

In conclusion, our study provides new insights into animal movement speed, activity index, and population density. However, the results should be interpreted with caution due to the limitations associated with the number of the CTs and the estimation methods used. Further research with more advanced tracking technologies and larger sample sizes would be beneficial to validate and expand upon our findings.

## ACKNOWLEDGEMENTS

We would like to thank Paula Aravena, Edmundo Schuster and Mario Talbotu for their help within our research.

### Funding

The authors received no funding for this work. The publication of this study was financially supported by the Open Access Publication Fund of the University of Veterinary Medicine Hannover, Foundation. The funders had no role in study design, data collection and analysis, decision to publish, or preparation of the manuscript.

### Grant Disclosures

The following grant information was disclosed by the authors:

Open Access Publication Fund of the University of Veterinary Medicine Hannover, Foundation.

## Competing Interests

The authors declare there are no competing interests.

## Author Contributions

- Oscar Skewes conceived and designed the experiments, performed the experiments, analyzed the data, prepared figures and/or tables, authored or reviewed drafts of the article, and approved the final draft.
- Annaluisa Kambas analyzed the data, prepared figures and/or tables, authored or reviewed drafts of the article, and approved the final draft.
- Paula Gädicke analyzed the data, prepared figures and/or tables, and approved the final draft.
- Oliver Keuling conceived and designed the experiments, authored or reviewed drafts of the article, and approved the final draft.

## Field Study Permissions

The following information was supplied relating to field study approvals (i.e., approving body and any reference numbers):

Field permit to conduct the study in the National Park of Villarrica was given by Ministerio de Agricultura (Chile), Depto. Areas Silvestres Protegidas Region de la Araucaria.

This camera trap study was authorized by the national forestry corporation CONAF in Chile. The Bioethics commission of the University of Concepcion approved the study and the pictures of people were handled following the guidelines of *Sharma et al. (2020)*.

## Data Availability

The raw data are available in the Supplementary Files.

## Supplemental Information

Supplemental information for this article can be found online at http://dx.doi.org/10.7717/peerj.18951#supplemental-information.

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
