# Peer review of "First wild boar density data from Araucaria forest in Patagonian Andes"

_PeerJ, doi:10.7717/peerj.18951_

## Round 0.1 · original submission · Major Revisions

Given the potential importance of your manuscript I am suggesting a revision. But, as noted, there are limitations to your study (number of traps, placement, etc) which need to be very clearly discussed along with what the resulting biases mean. Additionally, both reviewers identify a large number of areas that can be improved; in particular the flow and organization of your paper.

·

Basic reporting

This is the first study estimating wild boar density in endangered araucaria forest in the Patagonian Andes. Given the need for more evidence-based wildlife research in geographically understudied regions and the potentially detrimental impacts of wild boar on their ecosystem, I consider this study to be timely and important. Overall, the manuscript is well-written using scientific and unambiguous language and uses valid, state of the art methodolgy. However, the authors should pay attention to spelling out abbreviations that have not been introduced before, or may not be clear for a wider audience. Once introduced, they should be consistently using these abbreviations. In the Materials and Methods section, the reporting should be in the past tense. Sometimes, the authors have used the present tense instead. Some results that are reported come out of the blue (e.g. “We found significant differences in speed between the seasons (p value=0.0334, Kruskal-Wallis’s test).”). The authors should mention all statistical tests that they have carried out in the Materials and Methods section. The discussion should be more profound and grounded in literature to support all claims. I also miss a reflection on the study limitations (e.g. small sample size). Finally, I was not able to download the raw data, I beleive because it was not supplied. Please check this.

Experimental design

I have three major concerns about the experimental design:
- The sample size is very limited (only 10 CT deployements). I understand that financial, or possibly other, constraints may have limited the authors from deploying more CTs. However, they should at least discuss clearly the implications of low sample sizes on their analysis (mention your study limitations). Something that is entirely absent from this manuscript.

- If I understand it correctly, the way the authors present the density estimates obtained from REM is wrong. I suggest they have a look at the review of Gilbert et al. (2020). REM estimates a density for the collective viewshes of CTs, and not for individual CTs. This should be clarified in the manuscript. If I somehow misunderstood your analysis, others might too. So, please clarify.

- The Material and Methods section misses some critical information:
o Temporal unit of analysis (24-h periods, only nighttime periods (it seems so from the Results section))
o A brief (conceptual) explanation of REM, rather than just citing Palencia et al. (2021). Personally, I would also present the mathematical formula for REM, but I leave it up to the authors to decide if they include it or not.

Validity of the findings

The research objective is clearly stated: “estimating for the first time wild boar density in Araucaria araucana forests”. The methods by which the authors assess density are valid and widely adopted (state of the art), however the way in which they present the REM and its parameters is not always correct. REM estimates a density for the collective viewshes of CTs, and not for individual CTs. Finally, I could not assess the statistical soundness of the data, since they where not provided yet (or at least I could not download them).

Additional comments

Abstract:
“Also, density data …”: In my opinion, this sentence can be removed it is not essential to setting the context for your work.
“However, there is no such data …, until now”: I would suggest you drop “until now” and start the next sentence with “Hence, we monitored …”.
“Since we had strikingly different monthly encounters, we assume that the population is migrating to a lower altitude between the seasons.”: I guess this might follow later on in your manuscript, but the reason why you assume this migration pattern should be clarified and if possible your claim should be backed by literature.
“Since we had strikingly different monthly encounters, we assume that the population is migrating to a lower altitude between the seasons. Another indication for that could be the high daily range, which is also linked to the low food availability, as they have to move a lot more while searching for edibles.”: This reads almost as a discussion, the authors should try to remain factual and discuss the implications of their findings in the abstract.

Introduction:
L39: Please provide references for detrimental effects on biodiversity, agriculture, and livestock.
L44-55: Your line of reasoning is clear. ASF led to an increased interest in density estimation of wild boar in Europe, and, it turns out that density values vary considerably between different environments. However, the manuscript would be more clear, if the authors mentioned this more concisely. Currently, this lengthy paragraph breaks to flow of the introduction.
L56: I suggest the authors try to connect the previous paragraph with this paragraph by drawing upon the similarties between Europe and SA, i.e., wild boar as a pest species that requires informed management.
L71-74: This sentence is difficult to comprehend due to its length. Try to break it up into multiple, shorter sentences. Each of them with a clear message.
L75: You may drop “big” here and the sentence will read the same.
L76: “It must be taken into account” can also be ommitted.
L77: Watch out, you use “therefore” which implies that the information follows logically from the previous sentence, which is true for the absence of hunting, but not for negligible human disturbance. Modify accordingly.
L82-83: Do you have some specific hypothesis about the impact of wild boar on the Araucaria araucana forests.
L83: aims to generate/ aims to provide. A. araucana forests (Araucaria araucana was already introduced).
L86: The strength of the argument here is weak, i.e., you expect similar densities because the species is adaptable. But one could imagine plenty of situations in which wild boar display a similar degree of adaptability, and yet, have completely different population sizes. For instance, differences in hunting pressure, presence of predators, time since colonisation, history of diseases, … may all contribute to different densities between Europe and SA. I am not saying the assumption itself is false, but it should be better supported by strong arguments why you think it would be true. If not, I suggest that the authors just omit this statement from their manuscript.
Materials and Methods:
L92: km2 change to km2
L100: to avoid any confusion I would repeat “annual”, hence “mean annual temperature is 9.3 °C.
L102: “for approximately 45 days” the actual number of days with snow cover may vary from year to year.
L111: In which months do the Araucaria araucana trees bear fruits? What about other nut- or fruit-bearing trees? Briefly, discuss interannual variation in food availability.
L111-112: “The number of deployments …”: I understand the authors’ message, however I think this sentence should be rephrased such that it improves grammatically.
L113: CT locations … distance between camera’s.
L114: “Next, the points … in the field”: Not really necessary, you may omit this sentence.
L116: Spell out “diameter”.
L117: The authors use “CT’s”, “cameras” and “CTs” interchangeably. I would suggest to pick one and use it consistently. I often go for “CTs”, but that’s my personal preference 
L122: I would recommend to briefly state what is in the guidelines of Sharma et al. (2020).
L125: Again, I think it is usefull that you devote a few sentences to explaining the essence of the method described in Palencia et al. (2021). Not all readers have the time or motivation to look up these methods.
L126: R package ‘activity’ or R package activity
L127: I don’t think this is correct: “The REM density was calculated for each CT and then averaged for season.” REM estimates a density fort he collective viewshed of ALL CTs… This can be done using the entire dataset or for data of each year seperately. Clarify, what exactly it is that you did.
L128: Drop “statistically different”, this is a result. Also spell out “reps” as “replications”.
L129: “at our CTs (Cusack et al. 2015).”
L136: “measured”
L138: “reacted” and “CT”
L140: “n°”: I think you did not introduce this abbreviation yet. Please do or spell out: “number of”.
Results:
L146: It is not obvious from the Methods section that you only consider trapping nights. Make sure to clarify this in your “Material and Methods”.
L147: Spell out “radians”.
L150: “We found significant differences in speed between the seasons (p value=0.0334, Kruskal-Wallis’s test).” This comes out of the blue. Mention the Kruskal-Wallis test in the Materials and Methods section.
L157: Clarify what you mean by “cold season” and “warm season”.
L169-170: If I understood it correctly, your study area is 15 km2. Estimated densities are 2.7 ind./ km2 and 0.8 ind./ km2 for warm and cold seasons respectively. This makes for a huge difference in total density: +/- 40 individuals in the warm season Vs. 12 individuals during the cold season. While seasonal variation in density can be expected, this seems somewhat grotesque. Personally, I think the model is capturing variation in detectability/ movement patterns between seasons. Anyway, what I suggest you do, is to reasses whether these densities make sense to you, and if so, come up with a biological reasoning that you discuss in the “Discussion” section. If not, also discuss why your model may have estimated these different densities.

Discussion:
There is a few things that I miss from this discussion:
- A discussion on the sample size, given that it is suboptimal (in a sense that you would generally like to have at least around 30 deployments). Please elaborate on the implications of low sample size on your study results, and other study limitations.
- A more profound discussion on seasonal differences and how they might have impacted parameter estimates of the REM.
- Discussion on the precision of estimates, the authors discuss density estimates without mentioning the SEs/ CVs associated with them.
- Information from the literature on a possible carrying capacity of wild boar populations in araucaria forest/ Patagonian Andes. Or else some information supporting the notion that wild boar are a great danger to the forest regeneration.
Finally, the authors should try to be more consistent with their use of language, abbreviations etc.
L180: 1.6 individuals/ km2
L183: individuals/ km2  be consistent
L186: I follow your reasoning here, but is there any objective evidence that wild boar feeds on the Araucaria fruits? If not, their presence may not be of great relevance to wild boar. Please discuss.
L187: “This can presumbably …” It is not very obvious to what you are referring here. I beleive it is the density estimates of 1.6 individuals/ km2, and its discrepancy with higher European wild boar densities. But please clarify this.
L191-202: Great! This is a very clear and thoughtfull discussion of your specific study system. However, I miss a discussion on how all of this may have impacted the estimated parameters from your REM. You mention differences in movement capacity between seasons… So do you see this reflected upon the day range? What about density estimates, how are they influenced by this? Also, see my earlier comment on L169-170.
L203-204: I miss a reference here.
L209-210: Can you find any reference for this?
L214: Here you have it  You might want to use this reference in the M&M as well.

Conclusions:
L222-223: While a Discussion is there to, well, discuss… This seems rather speculative. What are your basis for considering the wild boar density too high for your ecosystem. This should be clearly stated in the discussion. If not, omit this sentence from the conclusions.
L229-230: Idem. I would suggest the authors to, at least, try to find some literature on the ecosystems carrying capacity. Or else, I advise that they include some calcultions/projections of a (roughly) estimated carrying capacity and how they arrive at it.

Figures and tables:
Table 2 – caption: “standard errors”
Figure 1: There is no clarification on the black circles indicated on the map. Please also clarify your figure’s legend in the caption as well as the inset maps.
Figure 3 – caption: It is a bit confusion that you metion: “from May 2020 until April 2022”, since you show aggregated trapping rates. I suggest you use stacked bins, which are color coded (or using different black-white pattern fills) according to the year for each month (Jan.-Dec.). This also allows readers to judge interannual differences in trapping rate.

Reviewer 2 ·

Basic reporting

The writing often lacks clarity, and several sections are poorly organized. In particular, the introduction lacks coherence as the topics seems disjointed and poorly connected.
The numerical data presented appears to have inconsistencies.
The figures generally lack sufficient explanations, and some contain labeling errors or citations unrelated to the text.

Experimental design

The original primary research is within the aims and scope of the journal.
Research question generally well defined, but the introduction needs to be organized.
REM is a well-established density estimation model, but there are major concerns with the authors' sampling design.
Methods are generally well described but need additional details.
It seems like this study is conducted in conformity with the prevailing ethical standards in the field.

Validity of the findings

Although all underlying data have been provided, there are concerns about whether the data obtained sufficiently reflect the characteristics of the wild boar population in study area.
Validity of the estimated wild boar density needs to be discussed in relation to the survey method.

Additional comments

I think this is a significant study that provides an estimate of the density of wild boar in Chile, where information on density is scarce. Seasonal changes in density in the study area are also interesting results. However, a major revision of manuscript is needed before it can be accepted for publication.

Major comments:
1) I’d suggest rechecking the flow of the story in your manuscript. The topics seems disjointed and poorly connected in the introduction. In my opinion, it would be better to first discuss the general impact of wild boars, followed by the importance of density estimation and the methods used for it. Then, it would be appropriate to explain the situation of wild boars in South America and the necessity of density estimation in the study area.

2) A rigorous survey design is required to estimate population density based on REM theory. In particular, the number of cameras, model selection, and installation procedures have a tremendous impact on estimation accuracy.
Only 10 cameras were installed in this study. But Rowcliffe et al. (2008) suggested that an absolute minimum of 20 camera locations should be deployed.
Rowcliffe et al. (2016) and Palencia et al. (2021) recommended to use camera traps with reliably fast trigger and photo burst rates to generate more accurate registration of animal trajectories inside the FOV, and in consequence, more accurate density results. I think Acorn 6210 is underspecified for this purpose. For example, Yajima and Nakashima (2021, https://doi.org/10.3106/ms2020-0055) reported that the Bushnell camera captured the dog 96% of the time, while the Ltl-Acorn camera missed about half of his passes. In fact, only 280 of the 370 encounters were used to estimate the speed of movement, and only one image was obtained in about 25% of the encounters in this study. In addition, there will be moves that were detected but not recorded. These mean that faster movement is not reflected in the estimation.
Why were cameras placed at 1 m above the ground? I believe the shoulder height of wild boars is about 80 cm, and that of piglets is even lower. Thus, at a close distance from the camera they would be able to evade the infrared sensor.
Therefore, I believe that these biases in the authors' density estimates by REM should be carefully discussed.

3) Overall, I think the manuscript has not been scrutinized. I have noted in minor comments what I have noticed, but I hope you will check the manuscript carefully before resubmitting it.

Minor comments:
L50: You defined it in L49, so just write indiv/km2.
L53-54: I think it's "/", not "*".
L67: For three or fewer authors, list all author names (e.g. Smith, Jones & Johnson, 2004).
L75: You write here 2500 ha, but 15km2 in L92. Which is the correct value? And the unit should be unified to km2.
L83: A. Araucana
L92: 1,200 to 1,400
L92: You write here 15km2, but 2500 ha in L75.
L100: 1,081 mm
L103: Why is Figure 2 cited for this sentence?
L110: How often did you maintain your camera traps?
L111: You write here March 2020, but May 2020 in L158 (Fig3). Which is the correct?
L112: Ltl, not Little.
L113: Is there a tool in Google Earth to randomly select points? If the authors manually selected the points by looking at the map, I would think that it is not a truly random. And why was only one point placed at a different location?
L113: 1,000-1,400
L115: Is such an environment with 10 m of clear vision in front of the camera lens common in the study area? If not, the estimated densities from the selected environment are biased and are not representative of the study area.
L116: I would like you to describe why the camera was mounted facing north or south.
L125: The abbreviation “REM” is not defined in the text so far.
L125: Palencia et al. (2021) is not listed in the References.
L127: Division of season is not defined anywhere in the method. What month was classified as which season?
L145: I think the total number of wild boar encounters should be written in the text.
L146-147: Season total (2454+2187) do not match total effort (4703). Furthermore, it does not match table 2.
L155: Figure 3 shows the lowest encounter rate in September, not August.
L156-157: I don't understand why the overall average is 2.0 when the average for each season is 1.9. And the SE values are different from Table 2.
L158: encounter, not trapping.
L158: I think “y” and “t” are not defined.
L158: You write here May 2020, but March 2020 in L111. Which is the correct?
L165: What is "e".
L169-170: You defined "indev/km2" in L49.
L176-177: I don't think this is a meaningful result. I think it is more important to note that there is variation in the estimated density between cameras and the result that there is a large variation in the warm season (Figure 4).
L180: You defined "indev/km2" in L49.
L181-183: I could not understand why they could not be compared. I think the authors should consider differences in estimation methods and compare it as well.
L190: I would like you to carefully discuss the estimated densities, not only environmental factors, but also biases derived from the survey design.
L218: Skotak, Drimaj & Kamler (2021) is not listed in the References.
L184&L222: I don't think it makes sense to compare the estimated density in this study with the European averages estimated under various environmental conditions. As you indicated in L49 there is a wide range in wild boar densities in Europe.
L225-227: Then I think you should make a comparison of your result with densities estimated under as similar conditions as possible.
L228: I think it would be "not that higher densities", not "not that lower densities".

References:
List all author names (L242, L274, L331).
Lack of “&” in the list of authors (L263, L319, L344).
Lack of indent (L266-268, L320)
Lack of pages (L270, L279, L296).
Lack of volume, issue, and pages (L305).

Table2:
I think “encounters/CT day” is correct, not “ind/CT day”.
The denominator and numerator are reversed in the y/t entries for cold and warm season.
Season total (2516+2181) do not match total effort (4703). Furthermore, it does not match L146-147.
I think “r [km]” is correct, not “r [m]”
95% Conf. interval values do not indicate an interval.

Figure 1:
I believe the longitude and latitude are appropriate, although a number with an unknown CRS is listed.
What do the four black circles indicate?

Figure 3:
I think “encounters” is correct, not “trapping”.
Vertical axis incorrectly lists minority points with commas.
Vertical axis is delimited by halfway numbers that are not at regular intervals. It should be changed to a scale of 0.05 increments.

Figure 4:
There is no explanation for the box plot. For example, is the bold line showing the median or the mean?

---

## Round 0.2 · Major Revisions

Both reviewers note that the manuscript has improved, but both also note that some of their previous recommendations were not dealt with well. In fairness to the reviewers, you should pay careful attention to the issues raised. If there is not truly substantial improvement in the next version, I will be forced to recommend rejection. Be sure to make the corrections in the computations, and provide working code (but do not ignore other comments).

·

Basic reporting

The writing still lacks clarity and consistency at times. For instance abbreviations are not used consistently throughout the text, or not introduced properly. Some sentences are too long and therefore hard to understand. The flow of the introduction, despite some modifications, still needs to be improved. The reporting of figures and numerical data improved. However, the color-scheme in Figure 1 needs some adjustments such that lakes are more distinctive from other land-classes. It would be good to also highlight other major vegetation types/ land-classes on the map. I would also omit Figure 4 from the manuscript entirely. It only causes unnecessary confusion by introducing density/CT out of nowhere (not mentioned in M&M).
Finally, the provided R-scripts do not support the reproducibility of the work presented in this article. An excel-file, containing the CT-data is supplied, but, it is not clear how this data should be imported into R. In summary, the authors should add some lines of code to their R-script initial lines that allow readers to reproduce their results.

Experimental design

As mentioned in an earlier round of review, the methods presented in this work are valid and well-established. Limitations concerning sampling design are now discussed. Nevertheless, the sample size (10 CTs) remains close to or beyond the absolute minimum number of CTs needed to produce accurate densities, using REM. The authors should pay attention to emphasize this not only in the discussion, but also in the conclusion.

Validity of the findings

Although all underlying data have been provided, I could not check the validity of the findings.
This is due to an ambiguous R-script. The authors should ensure reproducibility by adding a few lines of codes that imports their CT data and creates the right subsets (cold vs. warm season) of data for consequent modelling.

Additional comments

I have highlighted my additional comments in the attached pdf.

Reviewer 2 ·

Basic reporting

The manuscript has improved significantly, but I think that the authors' understanding of the REM is lacking. Additionally, there are still numerous errors in the numerical data and citations, which undermines its reliability. In my previous comment, I asked the authors to carefully check the manuscript before resubmitting it, but it appears that this was not done.

Experimental design

Although it is still insufficient, I think the biases arising from the experimental design are generally described in the discussion.

Validity of the findings

Based on the formula described in the methods section, I have concerns about the authors' understanding of REM and the resulting estimates.

Major comments:
1) The formula given in L132 is incorrect. If the authors used this formula to perform their estimation, the results would not show the correct density.
This formula is likely derived from eqn 1 in Rowcliffe et al. (2008). However, eqn 1 assumes a circular detection area and should not be directly applied to cameras with segment-shaped areas. The authors simply converted the radius “r” from eqn 1 into a detection zone “k”, but “2r” in eqn 1 represents the diameter of the circle, or the width of the covered path for a circular area. Rowcliffe et al. (2008) defined the width of the covered path for cameras in eqn 2, and subsequently proposed eqn 3 and 4 based on this. The authors should have used eqn 4.

2) The accuracy and consistency of the values shown below should be resolved.
L165-166: I don't understand why the overall average is 2.0 when the average for each season is 1.9. And the SE values are different from Table 2. I pointed out these issues last time, and you responded that it has “changed now”, but nothing has changed.
L171-172: These values are inconsistent with Table 1. And authors have not rounded off the numbers correctly.
L190: I have no idea what the values shown here were.

3) The following insufficient discussions need to be addressed.
L200-203: The detection angle and radius of the camera are considered in the REM, so the notion that they affect the estimation is inaccurate. Instead, you should focus on the camera's trigger speed and the height at which it is installed.
L203-204: The result for Palencia et al. (2021) is very interesting, but they mounted the cameras at heights of 30, 45, and 60 cm above the ground, which does not support your results for the camera mounted at 1 m above the ground.
L205-206: While it’s possible that many hares were photographed, this does not demonstrate a high detection rate for your cameras. For instance, being photographed 100 times out of 100 opportunities is not the same as being photographed 100 times out of 200 opportunities.

Additional comments

I commented for the PDF file, because the lines of the resubmitted PDF and DOCX were different.

Minor comments:
ALL: I think it would be better to standardize on standardize on one of the following: CT’s, CT´s or CTs.
L3: You typed 1 with a superscript and 2 with a special character.
L28: The correct spelling is “Barrios”.
L41: Need space before “In”
L44: Enetwild-Consortium et al. 2018 is not listed in the References.
L56,71,72: Scientific names (A. araucana) should be written in italics.
L64,78: The 2 in km2 should be typed as a superscript.
L82-83: QGIS is explained twice.
L91: Why the legends of figure 2 is here?
L99: avail“a”bility is correct.
L107: Apps & Mc Nutt 2018 is not listed in the References.
L127: This sentence (For the Random Encounter…) is repeated in the following sentence.
L148-149: This formula is difficult to understand, and it doesn't correspond to the explanation in L133-134. I believe it should be stated as follows: The encounter rate is defined as y/t.
L152: Di Renzo “et al.”, 2020 is correct.
L159-160: Seasonal classification should be defined in Material & Methods
L165: Two periods are typed.
L166: Why is Figure 3 cited for this sentence?
L179: I pointed out Figure 4 in the last time. I cannot find these values from Figure 4.
L188: Burton et al. 2015 & Kays et al. 2020 are not listed in the References.
L192: Massei et al. 2017 is not listed in the References.
L193, 197: Guerrasio et al. 2022 is not listed in the References.
L279: Change “and” to “&”, and “S. A. Ballari” to “Ballari S. A.”.
L298: Lack of “&”
L310: Enetwild-Consortium et al. 2023 is not quoted in the main text.
L326: Gürtler et al. 2023 is not quoted in the main text.
L343: Keuling et al. 2018 is not quoted in the main text.
L347: Kmetiuk et al. 2023 is not quoted in the main text.
L350: La Sala et al. 2023 is not quoted in the main text.
L364: Lack of “&”

Table2
L2: I think “camera trap days” is better than “camera traps*days”.
95% Conf. interval values do not indicate an interval.

---

## Round 0.3 · accepted · Accept

I am happy with the current version and the authors have satisfactory dealt with the comments. This is now ready for publication.